# Mapping quantum state dynamics in spontaneous emission

M. Naghiloo[1], N. Foroozani[1], D. Tan[1], A. Jadbabaie[1] & K.W. Murch[1,2]

The evolution of a quantum state undergoing radiative decay depends on how its emission is detected. If the emission is detected in the form of energy quanta, the evolution is characterized by a quantum jump to a lower energy state. In contrast, detection of the wave nature of the emitted radiation leads to different dynamics. Here, we investigate the diffusive dynamics of a superconducting artificial atom under continuous homodyne detection of its spontaneous emission. Using quantum state tomography, we characterize the correlation between the detected homodyne signal and the emitter's state, and map out the conditional back-action of homodyne measurement. By tracking the diffusive quantum trajectories of the state as it decays, we characterize selective stochastic excitation induced by the choice of measurement basis. Our results demonstrate dramatic differences from the quantum jump evolution associated with photodetection and highlight how continuous field detection can be harnessed to control quantum evolution.

[1] Department of Physics, Washington University, St Louis, Missouri 63130, USA. [2] Institute for Materials Science and Engineering, St Louis, Missouri 63130, USA. Correspondence and requests for materials should be addressed to K.W.M. (email: murch@physics.wustl.edu).

In spontaneous emission, an emitter decays from an excited state by releasing radiation into a quantized mode of the electromagnetic field. From the point of view of quantum measurement theory, the light–matter interaction entangles the quantum state of the emitter with its electromagnetic environment[1,2]. The emission field may therefore serve as a pointer state used to indirectly monitor the emitter's evolution. Subsequent measurements of the field convey information about the state of the emitter and consequently cause back-action[3]. Typically, spontaneous emission is detected in the form of energy quanta, resulting in an instantaneous jump of the emitter to a lower energy state. However, if the emission is measured with a detector that is not sensitive to quanta, but rather to the amplitude of the field, the emitter's state undergoes different dynamics over finite timescales. In this case, the emitter is predicted to diffuse through its state space rather than abruptly decay to the ground state in a quantum jump, and the choice of measurement on the pointer will affect the dynamics of diffusion[4–7].

In this work, we perform homodyne measurements of the spontaneous emission from an effective two-level system, formed by the strong light–matter interaction between a superconducting circuit and a microwave cavity. By performing phase-sensitive amplification, we selectively amplify and de-amplify orthogonal quadratures of the emission field, enforcing a choice of measurement basis on the pointer state and limiting the measurement back-action experienced by the emitter[8,9]. We use quantum state tomography, in addition to weak measurements, to study the back-action of homodyne measurements and track the time evolution of the emitter's state under radiative decay. Contrary to the evolution expected for photodetection, we observe quantum trajectories[3,8,10] that stochastically diffuse through the state space of the emitter. For certain phases of homodyne detection we observe that the emitter evolves towards its excited state under radiative decay[5]. This stochastic excitation demonstrates how phase-sensitive amplification of spontaneous emission can be utilized to control the emitter's evolution.

## Results

**Experimental set-up.** Our system (Fig. 1a) consists of an effective two-level emitter formed by the resonant interaction of a transmon circuit[11] and a three-dimensional waveguide cavity[12]. The strong light–matter interaction between the circuit and the cavity strips them of their individual character and gives rise to hybrid circuit-cavity states. We use the lowest energy transition ($\omega_0/2\pi = 6.83$ GHz) as an effective two-level system. Deliberate coupling to a 50 $\Omega$ transmission line results in a radiative decay rate $\gamma = 2.3 \times 10^6\,\mathrm{s}^{-1}$. The process of emission is described by the interaction Hamiltonian, $H_{\mathrm{int}} = \gamma(a^\dagger \sigma_- + a\sigma_+)$, where $a^\dagger$ ($a$) is the creation (annihilation) operator for a photon in the transmission line, and $\sigma_+$ ($\sigma_-$) is the pseudo-spin raising (lowering) operator. This interaction couples an arbitrary field quadrature $a^\dagger e^{i\phi} + a e^{-i\phi}$ to a corresponding emitter dipole $\sigma_- e^{i\phi} + \sigma_+ e^{-i\phi}$. Due to the Heisenberg uncertainty relations, the emitted radiation exhibits quantum fluctuations in its quadrature amplitudes. If these fluctuations are measured, they provide information on the emitter state and drive its stochastic evolution. Conversely, if the fluctuations are de-amplified, their information is no longer available, eliminating the corresponding stochastic back-action on the emitter state.

To accurately detect these quantum fluctuations, we perform phase-sensitive amplification[13] of outgoing signals near the emission frequency using a near-quantum-limited Josephson parametric amplifier[14,15]. In this mode of operation, the amplifier squeezes the outgoing light along an axis in quadrature space

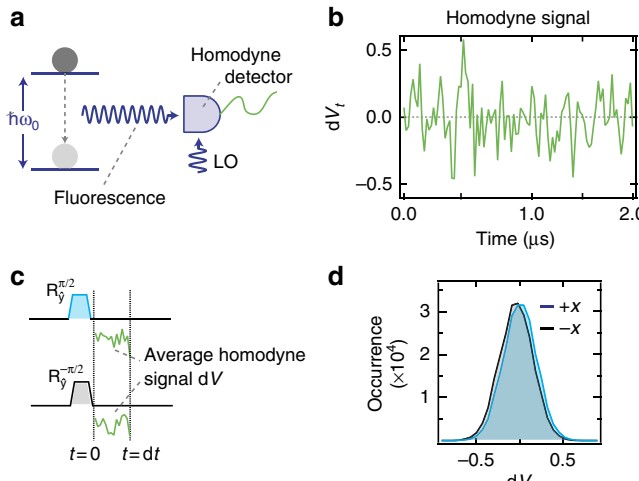

**Figure 1 | Experimental set-up.** (**a**) The experiment uses a near-quantum-limited Josephson parametric amplifier, labelled here as a homodyne detector, to perform homodyne measurements of the fluorescence emitted by an effective two-level system. (**b**) A sample of the dimensionless homodyne signal (denoted $dV_t$ at time step $t$). The noisy signal reflects the quantum fluctuations of the measured electromagnetic mode and is normalized such that its variance is $\gamma dt$. (**c**) To calibrate the measurement, we use $\pi/2$ rotations to prepare the emitter in the states $\pm x$ and average the ensuing homodyne signal for a time $dt = 20$ ns. (**d**) Histograms of the averaged homodyne signals **c** show how the measurement carries partial information about the $\sigma_x$ quadrature of the emitter's dipole. The strength of measurement, set by the emitter's decay rate $\gamma$, is inversely proportional to the overlap of the histograms.

given by the phase of the amplifier pump $\phi$. This constitutes a homodyne measurement of the amplified field quadrature $a^\dagger e^{i\phi} + a e^{-i\phi}$. Due to the emitter-field interaction, the choice of $\phi$ effectively enforces a choice of measurement basis on the emitter. In our experiment, we choose the amplifier phase $\phi = 0$; the corresponding noisy homodyne signal (denoted $dV_t$, Fig. 1b) is then sensitive to the emitter dipole $\sigma_- + \sigma_+ = \sigma_x$.

The variance of the homodyne signal originates not only from the quantum fluctuations of the detected mode, but also from losses and added noise in the amplification chain. We account for this loss of information into the incoherent and dissipative classical environment with the quantum efficiency $\eta$. Meanwhile, we treat the quantum noise as a Weiner process; the fluctuations of the measurement signal $dV_t$ in an infinitesimal time step $dt$ are described by stochastic noise increments $dW_t$. Known as Weiner increments[3], these are zero-mean, Gaussian random variables with variance $dt$. To accurately reflect this stochastic and dimensionless nature of the homodyne signal, we scale $dV_t$ such that it has a variance $\sigma^2 = \gamma dt$, with the full measurement record given by $dV_t = \sqrt{\eta\gamma}\langle\sigma_x\rangle dt + \sqrt{\gamma}dW_t$.

To experimentally demonstrate that our homodyne detection scheme is sensitive to a single quadrature of the emitter's dipole, we prepare the emitter in a specific state, perform homodyne measurement with $\phi = 0$, and integrate the resulting signal (Fig. 1c,d). By repeating the measurement for several iterations, we can create histograms of the homodyne signal. We compare the resulting distributions for two state preparations, $\pm x$ (the positive or negative eigenstates of the $\sigma_x$ Pauli operator). The observed separation of the two histograms, $\Delta V = 2\sqrt{\eta\gamma}dt$, gives the quantum efficiency of our detection set-up as $\eta = 0.3$.

**Conditional dynamics of radiative decay.** We now study the conditional dynamics of the emitter's state under radiative decay.

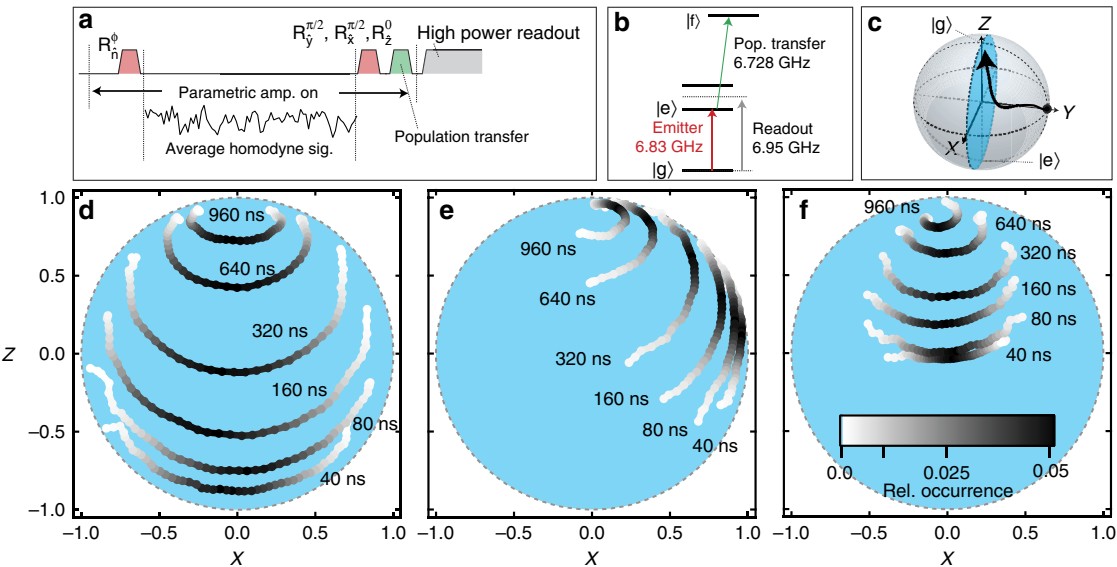

**Figure 2 | Mapping spontaneous decay.** (**a**) The experimental sequence prepares the emitter in an initial state, then homodyne detection is used to record the emitted radiation. Following a variable period of time, further rotations are applied to the emitter before state readout to perform quantum state tomography. To enhance the readout contrast, a pulse is applied to transfer the excited state population to a higher state of the system. (**b**) The level structure of the system and frequencies of the three microwave drives. The emitter is given by the lowest two energy levels. (**c**) We average the state tomography to determine $x \equiv \langle \sigma_x \rangle |_{\bar{V}}$, and $z \equiv \langle \sigma_z \rangle |_{\bar{V}}$ conditioned on the outcome of the homodyne measurement. These correlated tomography results are displayed on the $X$–$Z$ plane of the Bloch sphere for three different initial states: $-z$ (**d**) $+x$ (**e**) and $+y$ (**f**). The colour scale indicates the relative occurrence of each measurement value. Note the different back-action dynamics between **e** and **f**, a result of phase-sensitive amplification.

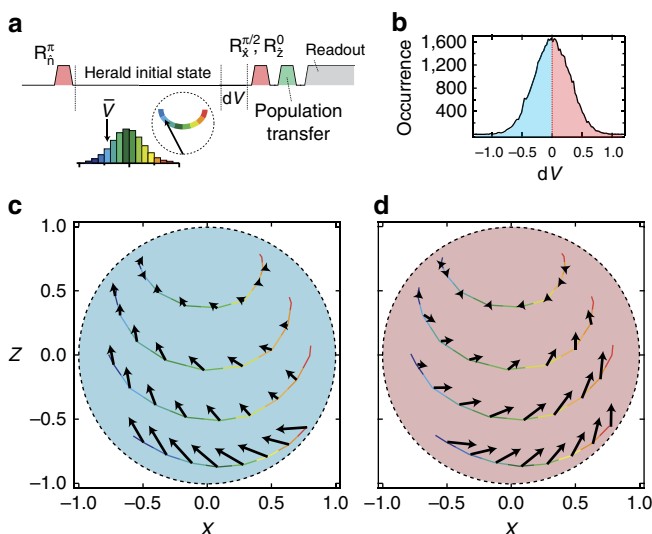

**Figure 3 | Back-action vector maps.** (**a**) An arbitrary initial state in the $X$–$Z$ plane of the Bloch sphere is prepared by heralding on the average homodyne signal $\bar{V}$. After heralding, we digitize the signal for an additional 40 ns to obtain d$V$. Finally, we use quantum state tomography to examine the conditional back-action based on the value of d$V$. (**b**) Histogram of the signals d$V$ which we separate into positive or negative d$V$. The back-action imparted on the emitter for negative (**c**) or positive (**d**) values of d$V$ is depicted by an arrow at different locations in the $X$–$Z$ plane of the Bloch sphere.

We conduct the experimental sequence depicted in Fig. 2a,b; we first use a resonant rotation to prepare an initial state, then obtain the average homodyne signal $\bar{V}$ by integrating the detected homodyne signal for a variable period of time, and finally perform projective measurements to conduct quantum state tomography as described in the Methods section. The results of

these projective measurements are averaged conditionally on the integrated homodyne signal. This yields the conditional Pauli averages, $\langle \sigma_x \rangle |_{\bar{V}}$, $\langle \sigma_y \rangle |_{\bar{V}}$, $\langle \sigma_z \rangle |_{\bar{V}}$. In Fig. 2d–f we plot $\langle \sigma_z \rangle |_{\bar{V}}$ and $\langle \sigma_x \rangle |_{\bar{V}}$ parametrically on the $X$–$Z$ plane of the Bloch sphere for different integration times. We study the conditional evolution for three different state preparations.

When the emitter is prepared in the excited state (Fig. 2d), the $x$-component of the state develops a correlation with the average homodyne signal. This highlights how our homodyne measurement provides an indirect signature[6] of only the real part of $\sigma_- = (\sigma_x + i\sigma_y)/2$. As the state is allowed more time to decay, it evolves to different deterministic arcs in the interior of the Bloch sphere.

Under phase-sensitive amplification, the choice of homodyne phase can vary the stochastic back-action on the emitter's state. To study this, we compare two different state preparations, $+x$ and $+y$. When the emitter is prepared in the state $+x$ (Fig. 2e), we observe that some of the conditioned states evolve towards the excited state[5]. This stochastic excitation is unique to amplitude measurements of the field quadrature, since such excitation is not possible under photodetection[6]. In contrast, when the emitter is prepared in the state $+y$, an eigenstate of the imaginary part of our measured operator $\sigma_- = (\sigma_x + i\sigma_y)/2$, the emitter dipole corresponds to the de-amplified quadrature of the emission field, and no stochastic excitation is observed (Fig. 2f). This different state preparation is equivalent to preparing the emitter in the same state $+x$ (as depicted in Fig. 2e) and instead changing the homodyne phase by $\pi/2$. This demonstrates how the choice of homodyne measurement phase can be used to control the evolution of the emitter.

We take advantage of the deterministic evolution of the emitter, conditioned on the integrated homodyne signal, to characterize the back-action at different points in the Bloch sphere. Figure 3 shows a vector map of the state evolution due to a specific detected homodyne signal d$V$ at various points. By preparing the emitter in the excited state and averaging the homodyne signal for various periods of time, we can prepare a

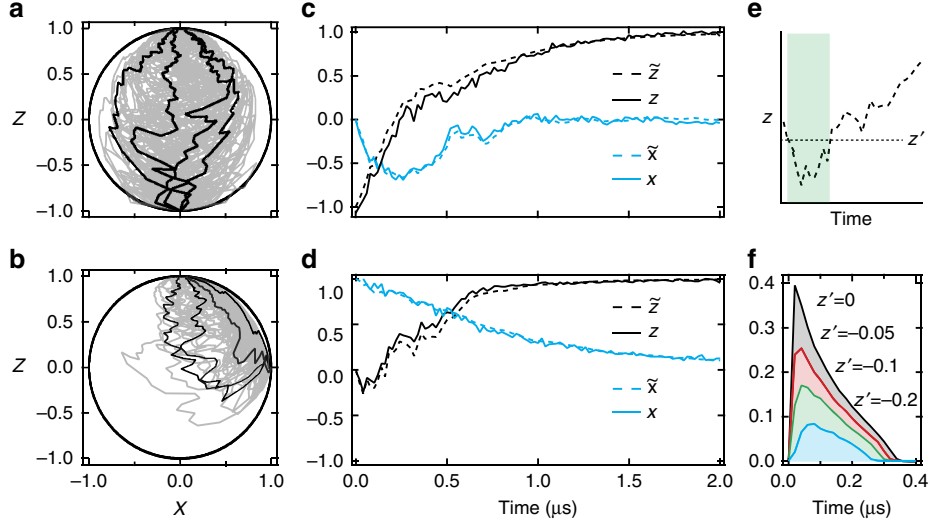

**Figure 4 | Quantum trajectories. (a,b)** Quantum trajectories of spontaneous decay calculated by the stochastic master equation, initialized at $-z$ **(a)** and $+x$ **(b)**. Several trajectories are depicted in grey, and a few individual trajectories are highlighted in black. **(c,d)** Individual trajectories $(\tilde{x}, \tilde{z})$ that originate from $-z$ **(c)** and $+x$ **(d)** are shown as dashed lines and the tomographic reconstructions (Methods section) based on projective measurements are shown as solid lines. **(e)** For trajectories that are initialized along $+x$ some are excited (attaining values below a threshold $z'$). **(f)** Fraction of the trajectories that are excited below the threshold $z'$ versus time.

nearly arbitrary mixed state through heralding. After selecting a decay time and a specific initial state $(x_i, z_i)$, based on an average signal $\bar{V}$, we digitize the homodyne signal for an additional time $dt = 40$ ns to obtain $dV$. We then use quantum state tomography to determine the final state $(x_f, z_f)$, conditioned on the detection of $dV$ within a specified range. The back-action at a specific location in state space, associated with the detection of a given value of $dV$, is provided by the vector connecting $(x_i, z_i)$ and $(x_f, z_f)$. The back-action vector maps demonstrate how positive (negative) measurement results push the state towards $+x$ $(-x)$. Furthermore, the maps show that the back-action is stronger near the state $-z$, indicating that the measurement strength is proportional to the emitter's excitation.

**Quantum trajectories**. The back-action maps that we present in Fig. 3 allow us to calculate the evolution of the emitter's state conditioned on a sequence of homodyne measurement results. Formally, this evolution is described by the stochastic master equation[5],

$$d\rho = \gamma \mathcal{D}[\sigma_-]\rho\, dt + \sqrt{\eta\gamma}\,\mathcal{H}[\sigma_-\, dW_t]\rho. \quad (1)$$

Where $\mathcal{D}[\sigma_-]\rho = \sigma_-\rho\sigma_+ - \frac{1}{2}(\sigma_+\sigma_-\rho + \rho\sigma_+\sigma_-)$ and $\mathcal{H}[O]\rho = O\rho + \rho O^\dagger - \mathrm{tr}[(O + O^\dagger)\rho]\rho$ are the dissipation and jump superoperators, respectively. When we ignore the results of homodyne monitoring (for example by setting $\eta = 0$), the state follows deterministic evolution from an initial state to the ground state, as described by the first term of equation (1). The second term accounts for information conveyed by the homodyne measurement through stochastic noise increments $dW_t$. We can recast this stochastic master equation in terms of the Bloch vector components $x, z, y$,

$$dx = -\frac{\gamma}{2}x\, dt + \sqrt{\eta}\left(1 - z - x^2\right)\left(dV_t - \gamma\sqrt{\eta}\,x\, dt\right), \quad (2)$$

$$dz = \gamma(1 - z)\, dt + \sqrt{\eta}\,x(1 - z)\left(dV_t - \gamma\sqrt{\eta}\,x\, dt\right), \quad (3)$$

$$dy = -\frac{\gamma}{2}y\, dt - \sqrt{\eta}\,xy\left(dV_t - \gamma\sqrt{\eta}\,x\, dt\right). \quad (4)$$

We now turn to calculating individual quantum trajectories for the emitter's state. In Fig. 4, we prepare the emitter in the excited

state and then digitize the detected homodyne signal for $2\,\mu s$. Based on this signal, we use equations (2–4) to calculate the emitter's trajectory using time steps of $dt = 20$ ns. Instead of taking a straight path to the ground state, the trajectory diffuses through the Bloch sphere, subject to back-action from the measured quantum fluctuations of the emission field.

We also study quantum trajectories originating from the state $+x$. In this case, the stochastic back-action causes some of the trajectories to become more excited as they decay under homodyne detection. In Fig. 4f, we quantify this feature by extracting the probability of excitation above a certain threshold at different times. By examining the measurement term in equation (3), proportional to $\sqrt{\eta}$, we see that the state at $+x$ will be stochastically excited if the Weiner increment $dW_t$, obtained from the detected signal $dV_t$, is less than $-\sqrt{\gamma/\eta}\,dt$, predicting that $\sim 35\%$ of the trajectories should be excited in the first time step.

## Discussion

In recent years, several experiments have demonstrated control over the emission process by either altering vacuum fluctuations[16] or engineering the electromagnetic environment[17,18], allowing spontaneous emission to be used as a resource for quantum information[19,20]. In addition, the entanglement between a quantum emitter and its spontaneous emission field has been studied in experiments using natural atoms[1] and solid state systems[2], and can be used to herald entanglement between spatially separated systems[21]. Our work highlights how spontaneous emission can also be used as a resource for quantum measurement, where the emission field serves as a pointer system for indirect measurements of the emitter state. Contemporary experiments[7–10,22–26] that harness Bayesian statistics or use quantum optics to track the evolution of quantum states have yielded a deeper understanding of quantum measurement evolution. Here, we have shown how specific quadrature measurements of the fluorescence from a quantum emitter result in a rich conditional evolution of the state. We have harnessed this evolution to map out the back-action associated with such measurements, and we have tracked the individual quantum trajectories an emitter takes when decaying through

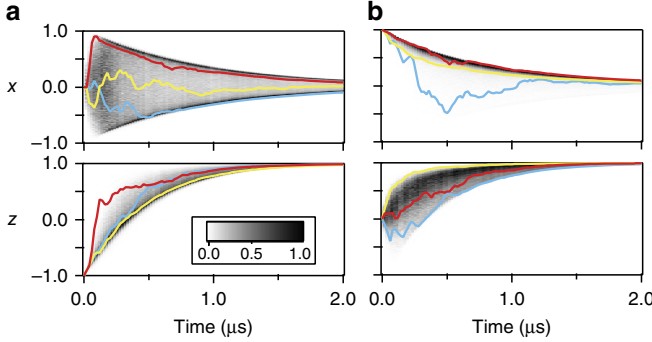

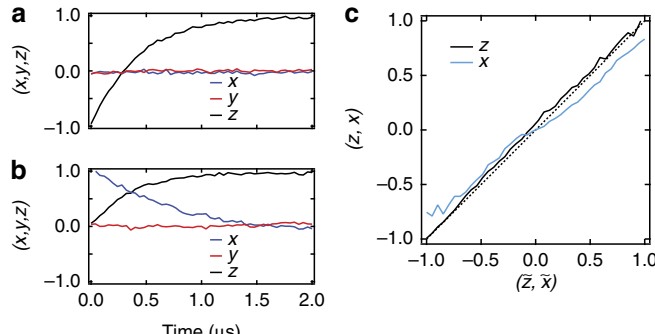

**Figure 5 | State histograms.** Greyscale histograms represent the distribution for values of $x$ and $z$ at each time point. The greyscale shading is normalized such that the most frequent value is 1 at each time point. (**a**) Histograms of the state when the emitter is initialized in the state $-z$ with a few sample trajectories shown in colour. (**b**) Histograms associated with decay from the state $+x$.

**Figure 6 | Tomography calibrations.** The ensemble decay as determined by projective measurements for initial states $-z$ (**a**) and $+x$ (**b**). (**c**) Tomographic validation for the ensemble of trajectories shows the average tomography values $(x, z)$ versus the values obtained from individual trajectories.

fluorescence. In contrast to the instantaneous dynamics of emission due to measurements of quanta, here we show that spontaneous emission may also occur over finite timescales.

Measurements, and more broadly, control over a quantum environment, can in principle be used to steer quantum evolution[27,28]. Though the physical mechanism of energy decay remains constant between homodyne detection and photodetection, by changing the basis for the measurement of the emission field, we have significantly altered the emitter state's dynamics. In this work, quantum noise of the pointer state drives the evolution of the emitter, demonstrated clearly by our use of a phase-sensitive amplifier. By squeezing the pointer state, we change the nature of its quantum fluctuations, and therefore cause selective measurement back-action on the emitter. Such control over the quantum light–matter interaction has the potential to advance techniques in fluorescence based imaging, and will be essential in quantum feedback control[3,9,29] of quantum systems.

## Methods
**Device fabrication and parameters.** The emitter system consists of a transmon circuit characterized by charging energy $E_C/h = 270$ MHz and Josephson energy $E_J/h = 24.6$ GHz. The circuit was fabricated by double-angle evaporation of aluminium on a high resistivity silicon substrate. The circuit was then placed at the centre of a waveguide cavity (dimensions $34.15 \times 27.9 \times 5.25$ mm) machined from 6061 aluminium. The cavity geometry was chosen to be resonant with the lowest energy transition of the transmon circuit. The resonant interaction between the circuit and the cavity (characterized by coupling rate $g/2\pi = 136$ MHz) results in hybrid states, as described by the Jaynes–Cummings Hamiltonian. The cavity is deliberately coupled to two $50\,\Omega$ cables: one weakly coupled port, characterized by coupling quality factor $Q_c \simeq 10^5$, is used to drive the system, while a more strongly coupled port $Q_c \simeq 10^4$ sets the total radiative decay time of the system. This configuration results in an effectively 'one dimensional atom', where all of the radiative decay is captured by the strongly coupled cable[16]. Spontaneous emission from this 'artificial atom' is amplified by a near-quantum-limited Josephson parametric amplifier, consisting of a 1.5 pF capacitor, shunted by a superconducting quantum interference device (SQUID) composed of two $I_0 = 1\,\mu$A Josephson junctions. The amplifier is operated with negligible flux threading the SQUID loop and produces 20 dB of gain with an instantaneous 3-dB-bandwidth of 20 MHz.

We used standard techniques to measure the energy decay time $T_1 = 430$ ns and Ramsey decay time $T_2^* = 830$ ns, indicating that the emitter experiences a negligibly small amount of pure dephasing. We also examined the equilibrium state populations of the emitter using a Rabi driving technique[30], and found the excited state population to be less than 3%.

**State tracking.** We use a master equation (equivalent to equations (2–4)) to propagate the density matrix for the emitter's state conditioned on the detected homodyne signal. The signal is digitized in 20 ns steps, and scaled such that its variance is $\gamma dt$. At each time step, we update the density matrix components $\rho_{11}[i]$ and $\rho_{01}[i]$ based on the detected measurement signal $dV[i]$, where $z \equiv 1 - 2\rho_{11}$ and

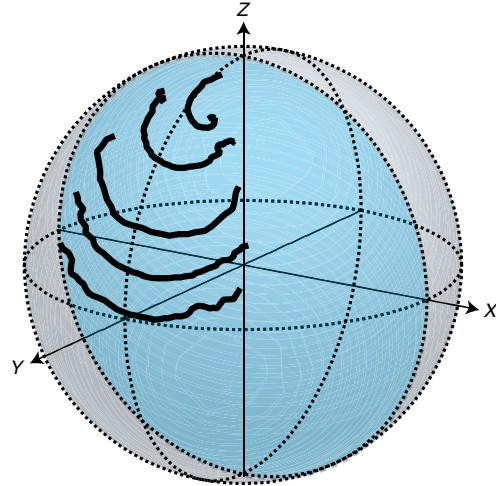

**Figure 7 | Spontaneous decay from the state $+y$.** The emitter's state at different times conditioned on the integrated homodyne measurement signal. The decay times are 80, 160, 320, 640 and 960 ns, and the data correspond to what is depicted in Fig. 2f. The $X$–$Z$ plane plotted in Fig. 2f is highlighted in blue.

$x \equiv 2\mathrm{Re}[\rho_{01}]$. Our state update is consistent with the Itô formulation of stochastic calculus.

$$\rho_{11}[i+1] = \rho_{11}[i] - \gamma\rho_{11}[i]dt \\ - \sqrt{\eta}(dV[i] - \sqrt{\eta\gamma}2\rho_{01}[i]dt) \\ \times (2\rho_{01}[i]\rho_{11}[i]) \tag{5}$$

$$\rho_{01}[i+1] = \rho_{01}[i] - \gamma\rho_{01}[i]/2dt \\ + \sqrt{\eta}(dV[i] - \sqrt{\eta\gamma}2\rho_{01}[i]dt) \\ \times (\rho_{11}[i] - 2\rho_{01}[i]\rho_{01}[i]) \tag{6}$$

**Ensemble dynamics.** Based on $9 \times 10^5$ repetitions of the experiment and associated quantum trajectories, we can examine ensemble dynamics of the paths on the Bloch sphere taken by our decaying emitter. The behaviour of single trajectories characterizes the dynamics of spontaneous decay subject to homodyne detection, and is distinctly different than the full ensemble behaviour that decays deterministically towards the ground state.

Figure 5 displays greyscale histograms of the state at different points in time for two different initial conditions. For trajectories initialized in $-z$ (Fig. 5a), these histograms demonstrate how the decay paths are restricted to a deterministic arc in the Bloch sphere. Curiously enough, a state prepared in a traditional eigenstate of spontaneous emission will develop some quantum coherence when monitored under homodyne detection. The $x$-components of such trajectories may be pinned to the edges of this arc on the $X$-axis, or instead may oscillate about the central

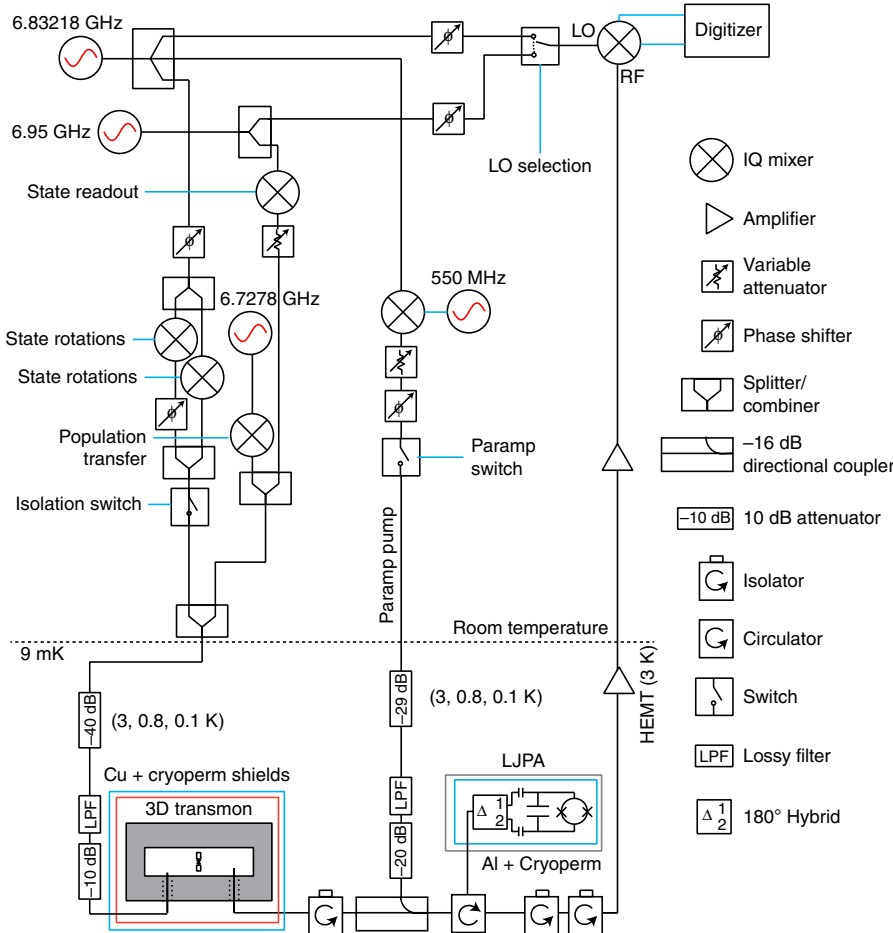

**Figure 8 | Experimental set-up.** A single microwave frequency generator is used for the qubit rotations, the amplifier pump and the local oscillator for demodulation. A second generator, operating at the high-power resonance frequency of the cavity is used for state readout for tomography. The third generator is used to transfer state population to a higher excited state of the circuit-cavity system to enhance the readout contrast.

value of $x = 0$. We note that though the trajectories exhibit an immediate diffusive behaviour for short timescales, the decay of coherence takes over at longer timescales, indicated by a decreasing upper bound on the stochastically acquired coherence. Examining behaviour along the $Z$-axis, we see that though some trajectories may decay by more quickly approaching the ground state, no trajectory may decay more slowly in $z$ than a specific lower bound at each time step.

On the other hand, when the emitter is initialized along $+x$ in a superposition of its excited and ground states, the histograms of the Bloch sphere coordinates show different behaviour (Fig. 5b). The $x$-component of the trajectory encounters a decreasing upper bound on its maximum value, once more illustrating motion along a shrinking deterministic arc. The $z$-component, however, can exhibit extremely varied behaviour. In addition to following the average decay path, the state may also stochastically excite, or it may rapidly decay in $z$ while approaching the surface of the Bloch sphere. Currently, it is these states that rapidly decay that have the highest purity on average, retaining the most information about the state. In comparison, due to our limited measurement efficiency, stochastically excited trajectories become more mixed as they diffuse towards the excited state. We note that for $\eta = 1$, all of our trajectories, regardless of dynamics, would describe pure states confined to move only on the surface on the Bloch sphere.

In fact, we expect the ensemble ratio of stochastically excited trajectories to increase with increasing $\eta$. As mentioned in the main text, trajectories experience $dz < 0$ when the Weiner increment obtained from the measurement record satisfies $dW_t < -\sqrt{\gamma}dt/\sqrt{\eta}x$. Recall that $dW_t$ is a zero-mean random variable distributed with variance $dt$, and consider the back-action experienced by trajectories initialized with $x = 1$. Naively, the probability of stochastic excitation is then given by the integral,

$$\int_{-\infty}^{-\sqrt{\gamma/\eta}dt} dW_t (2\pi dt)^{-1/2} e^{-dW_t^2/2dt}. \qquad (7)$$

As $\eta$ increases, so does the value of this integral. For $\eta = 1$ and a time step $dt = 20$ ns, the probability for stochastic excitation for our system reaches a maximum value of $\sim 41.5\%$. For our measured quantum efficiency of $\eta = 0.3$, we expect $\sim 35\%$ of trajectories to excite in the first time step.

**Tomography and readout calibration.** All tomography results are corrected for imperfect state preparation and readout fidelities. We perform state readout by first applying a resonant pulse at 6.73 GHz to transfer the excited state population to a higher excited state, and then proceed to drive the bare cavity resonance at 6.95 GHz at high power to conduct the Jaynes–Cummings high-power readout technique[31]. Tomography for $y$ and $x$ is achieved by first applying a 40 ns $\pi/2$ rotation about the $X$ or $Y$ axes. The combined state preparation and readout fidelity (80%) was determined from the contrast of resonant Rabi oscillations. Each experimental sequence includes separate calibration measurements used to determine the readout level of the ground state and the prepared excited state. These levels are used to scale the tomography results. Figure 6a,b shows the ensemble decay curves for the state preparations $-z$ and $+x$.

The emitter's state is characterized by expectation values $(x, z)$. To characterize accuracy of the state tracking, we compare the expectation values that are calculated for a single iteration of the experiment to the values obtained from an ensemble of projective measurements. In Fig. 4 we show this comparison to reconstruct an individual trajectory. To accomplish this, we denote an individual trajectory $(\tilde{x}(t), \tilde{y}(t), \tilde{z}(t))$ (Note that $\tilde{y}(t) = 0$). At each time point, we perform several experiments of total duration $t'$, followed by one of three tomography and readout sequences. For each of these experiments, we calculate $(x(t'), z(t'))$; if $x(t')$ and $z(t')$ are within $\pm 0.12$ of $\tilde{x}(t')$ and $\tilde{z}(t')$, then the subsequent tomography result is included in the tomographic validation at $t'$. We follow this process for each $t'$ along the trajectory, resulting in a tomographic reconstruction of the trajectory.

We can further test the predictions given by the individual trajectories for all runs of the experiment at all times. Figure 6c displays the average projective measurement outcomes conditioned on the values of $\tilde{x}(t')$ or $\tilde{z}(t')$ compared with the values $\tilde{x}(t')$ or $\tilde{z}(t')$ showing good agreement between the individual trajectories and the projective measurements.

**Phase-sensitive back-action.** When the emitter is initialized in $+y$ the state dynamics are not confined to the $X$–$Z$ plane. Figure 7 displays the state conditioned on the integrated homodyne signal and shows how the $y$-component does not

acquire a correlation with the measurement signal. This may be understood as a result of phase-sensitive amplification with $\phi = 0$. When we perform our homodyne measurement of the real part of $\sigma_-$, we de-amplify the quadrature containing information on the imaginary part of $\sigma_-$, corresponding to $\sigma_y$ on the Bloch sphere. The de-amplification of this orthogonal signal suppresses the magnitude of its quantum fluctuations, effectively eliminating the information associated with the $\sigma_y$ quadrature of the emitter's dipole. Therefore we do not perform weak measurements of $\sigma_y$, and we do not observe quantum dynamics such as stochastic excitation.

We may also understand this phenomenon by examining the $dz$ and $dy$ segments of the stochastic master equation provided in the main text. The presence of an $xy$ coefficient on the measurement term in equation (4), means the stochastic back-action has no effect on the state when it is in an eigenstate of $\sigma_x$ or $\sigma_y$, limiting dynamics to a deterministic reduction in $y$. Meanwhile, if we examine equation (3) after factoring out a common factor of $(1 - z)$, which serves to push the trajectory towards the ground state, we see the measurement term is proportional only to $x$. Therefore, for a state prepared with $y = \pm 1$, there will be no initial stochastic excitation, and the state will begin its decay by deterministically approaching the ground state. However, once fluctuations in the measurement signal cause the state to acquire a nonzero $x$ value, the trajectory's dynamics will cease to be trivial.

**Experimental set-up.** Figure 8 displays a simplified schematic of the experimental set-up. A single generator is used for qubit rotations, the amplifier pump, and demodulation of the amplified signal. The parametric amplifier is pumped by two sidebands that are equally separated from the carrier by 550 MHz, allowing for phase-sensitive amplification without leakage at the emitter's transition frequency. The experimental repetition rate is 8 kHz.

**Data availability.** The data that support the findings of this study are available from the corresponding author on request.

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

## Acknowledgements

We thank A.N. Jordan, K. Mølmer and P. Harrington for discussions. This research was supported in part by the John Templeton Foundation and the Sloan Foundation, and used facilities at the Institute of Materials Science and Engineering at Washington University.

## Author contributions

M.N., N.F. and K.W.M. performed the experiments and D.T. fabricated the samples. All authors contributed to the data analysis and writing the manuscript.

## Additional information

**Competing financial interests:** The authors declare no competing financial interests.

