## [Peer Review File · Nature Communications]

Transferred manuscripts:

Reviewers' Comments:

Reviewer #1 (Remarks to the Author)

I am very pleased with this new version of the manuscript prepared by the authors. I would like to thank them for considering my suggestions and for reviewing text, captions and figures accordingly in a very satisfactory way.

I would this time only suggest to remove two small typos in the caption of Figure 2:

" ... the relative occurrence of each" instead of "... the relative occurrence of the of the each"

" ... between (e), and (f) ..." instead of " ... between (e), and +y (f) ..."

I would strongly recommend the publication of the current manuscript in Nature Communication.

Reviewer #2 (Remarks to the Author)

The authors have remarkably improved the presentation of their manuscript, that is now much more accessible for a wider public.

The novel version of the manuscript, and the answer to referees, only mildly convinced me about the novelty of this work, when compared to the papers cited by myself and the other referee. Still my limited understanding of the subtleties of the experimental system could be to blame in this regard, and I thus consider myself content in this sense.

The only point I strongly object in this work concerns the use of the word "spontaneous emission". I consider the claim about this work "giving insight into spontaneous emission" to be misleading, as the emission is not spontaneous but subject to the back-action of the homodyne measurement. The emission is here "Spontaneous" only in the sense of "Not stimulated", but that's clearly not what the manuscript implies.

I invite the authors to change any occurrence of "spontaneous decay" and "spontaneous emission" in the title and in the text of the manuscript with "evolution subject to homodyne detection".

Reviewer 1

I am very pleased with this new version of the manuscript prepared by the authors. I would like to thank them for considering my suggestions and for reviewing text, captions and figures accordingly in a very satisfactory way.

I would this time only suggest to remove two small typos in the caption of Figure 2: "... the relative occurrence of each" instead of "... the relative occurrence of the of the each" "... between (e), and (f) ..." instead of "... between (e), and +y (f) ..."

I would strongly recommend the publication of the current manuscript in Nature Communication.

Response to Reviewer 1— We are happy that the reviewer gives a strong recommendation for publication. We have made the suggested corrections to the caption of Figure 2.

Reviewer 2

The authors have remarkably improved the presentation of their manuscript, that is now much more accessible for a wider public.

The novel version of the manuscript, and the answer to referees, only mildly convinced me about the novelty of this work, when compared to the papers cited by myself and the other referee. Still my limited understanding of the subtleties of the experimental system could be to blame in this regard, and I thus consider myself content in this sense.

The only point I strongly object in this work concerns the use of the word "spontaneous emission". I consider the claim about this work "giving insight into spontaneous emission" to be misleading, as the emission is not spontaneous but subject to the back-action of the homodyne measurement. The emission is here "Spontaneous" only in the sense of "Not stimulated", but that's clearly not what the manuscript implies.

I invite the authors to change any occurrence of "spontaneous decay" and "spontaneous emission" in the title and in the text of the manuscript with "evolution subject to homodyne detection".

Response to Reviewer 2— We are happy that the reviewer is content with publication of the manuscript. Regarding the reviewer’s invitation to change the occurrence of “spontaneous decay” and “spontaneous emission” we have given our use of the term “spontaneous emission” careful consideration. We use these terms in two ways in our manuscript: one is to describe the type of coupling of the emitter to the environment, and the other is to describe the radiative emission of the emitter that is detected. We believe that both usages are correct and respectfully decline the reviewer’s invitation to change all instances. We have however removed the claim that we “give insight into spontaneous emission”. Our reasons for keeping this language are detailed below.

First, spontaneous emission really is the correct description of the interaction of the emitter with the environment. The innovation of our work is that we use the emission that is released into the environment as a pointer state for quantum measurement, but we emphasize that if we did not make measurements of the emitted radiation, the emitter would still emit by coupling to free space and decaying to the ground state. In contrast to some of our previous work—where we can control the strength of the measurement and homodyne measurement also leads to interesting conditional evolution—in this case there is no way to turn off the physical mechanism for spontaneous emission and the emitter decays regardless of how the emission is detected. We believe it is important to use spontaneous emission because it highlights a key result of our work: how the same mechanism for evolution in quantum mechanics can result in different dynamics depending on how one performs a quantum measurement.

Second, we note that the term “spontaneous emission” is used in many other papers in the literature to describe this specific process. The main theoretical inspiration for the work, the paper by Bolund and Mølmer (Phys. Rev. A, 89:023827, Feb 2014.), specifically refers to the process as spontaneous emission in the first sentence of the abstract “We study the dynamics of an atomic two-level system decaying by spontaneous emission of light”. In fact, the stochastic master equation that we present is derived in this paper. The work by P. Campagne-Ibarq (Phys. Rev. X, 6:011002, Jan 2016) also refers to the process as spontaneous emission on multiple occasions “This illustrates that the measurement back-action associated with spontaneous emission is as strong as the qubit excitation is large”. The theory work by A. Jordan (arXiv:1511.06677, 2015.) refers to spontaneous emission in the title: “Anatomy of fluorescence: Quantum trajectory statistics from continuously measuring spontaneous emission.” Because the term spontaneous emission is used in several peer-reviewed theoretical and experimental works we feel justified in using the term spontaneous emission in our title and throughout the manuscript.

Kater Murch